# Immune Enhancement of Nanoparticle-Encapsulated Ginseng Stem-Leaf Saponins on Porcine Epidemic Diarrhea Virus Vaccine in Mice

**DOI:** 10.3390/vaccines10111810

**Published:** 2022-10-27

**Authors:** Fei Su, Lihua Xu, Yin Xue, Wei Xu, Junxing Li, Bin Yu, Shiyi Ye, Xiufang Yuan

**Affiliations:** 1Institute of Animal Husbandry and Veterinary Science, Zhejiang Academy of Agricultural Sciences, Hangzhou 310002, China; 2Zhejiang Center of Animal Disease Control, Hangzhou 310020, China; 3Zhejiang Provincial Key Laboratory of Preventive Veterinary Medicine, MOA Key Laboratory of Animal Virology, Center for Veterinary Sciences, College of Animal Sciences, Zhejiang University, Hangzhou 310058, China

**Keywords:** ginseng stem-leaf saponins, nanoparticles, mucosal adjuvant, porcine epidemic diarrhea virus

## Abstract

Porcine epidemic diarrhea virus (PEDV) causes severe enteric disease in pigs, particularly neonatal piglets. Current vaccines do not provide complete protection against PEDV. Ginseng stem-leaf saponins (GSLS), a promising oral adjuvant candidate, can improve intestinal immune responses in poultry and mice. However, its low stability limits further use. Poly lactic-co-glycolic acid (PLGA), a biocompatible and biodegradable nanoparticle, has been widely used in biomedicine for stable and targeted drug delivery. In this study, we developed GSLS-PLGA nanoparticles (GSLS-NPs) and evaluated the mucosal adjuvant efficacy in vitro and in vivo. GSLS-NPs significantly enhanced antigen internalization and pro-inflammatory cytokine secretion by DC2.4 cells. Mice orally administered GSLS-NPs before intramuscular inoculation generated CD11b^+^CD8α^−^ and CD11b^−^CD103^+^ dendritic cells in the spleen and draining mesenteric lymph nodes, respectively, which are the types mainly responsible for antigen presentation. Additionally, enhanced neutralizing and non-neutralizing antibody responses and expanded activities of specific effector and memory CD4^+^ and CD8^+^ T cells were also observed in mice immunized with PEDV vaccines plus GSLS-NPs compared to mice receiving the vaccines alone. Furthermore, GSLS-NPs showed a good safety profile and presented great advantages over GSLS aqueous solution. Collectively, our results highlight the potential of GSLS-NPs as a mucosal adjuvant and provide an attractive vaccination strategy for combatting PEDV. Further study is required to evaluate the efficacy of this mucosal adjuvant in swine.

## 1. Introduction

Porcine epidemic diarrhea virus (PEDV), a member of the genus *Alphacoronavirus* in the family *Coronaviridae*, causes acute enteric symptoms characterized by severe diarrhea, vomiting, and dehydration in pigs, causing high mortality in neonatal piglets that reaches nearly 100% [1]. In 2013–2014, a PEDV outbreak led to approximately 8 million pigs dying in the United States and cost between USD 900 million and 1.8 billion in annual losses [2]. PEDV is now considered one of the most devastating porcine viruses that has emerged or re-emerged and presents a significant threat to the worldwide swine industry [3]. The main PEDV transmission route is the fecal-oral route, although airborne transmission via the fecal-nasal route is also frequent within or between farms. As PEDV mainly infects and replicates in the villus epithelium of the small intestine, especially in the jejunum and ileum, locally produced virus-specific immunoglobulin A (IgA) antibodies are the most important factor in the primary adaptive defense at mucosal surfaces [4]. However, repeated outbreaks of PEDV on large farms and the emergence of highly pathogenic strains indicate that the effectiveness of current parenteral vaccination is not sufficient. Therefore, more effective vaccination strategies for PEDV prevention are urgently needed. One alternative solution is to develop potent adjuvants to improve mucosal immunity.

Ginseng saponins, consisting of steroid or triterpene aglycone linked to one or more sugars, are believed to be the major bioactive constituent of *Panax ginseng* C. A. Meyer. The total ginseng saponins extracted from the stems and leaves (GSLS) are composed of the ginsenosides Re, Rg1, Rb2, Rd and so on, which exhibit pharmacological activities similar to those from the roots and are available at a lower cost [5]. GSLS are widely used as an oral adjuvant for improving vaccination in poultry and mice. Zhai et al. reported that oral administration of GSLS could significantly enhance the efficacy of an infectious bursa vaccine in chickens, increasing the populations of intestinal intraepithelial lymphocytes (iIELs) and lamina propria IgA-secreting cells from the duodenum, jejunum and ileum mucosa when compared with the vaccine alone, and provide more effective protection against the virulent strain of chicken infectious bursal virus [5]. In addition, oral administration of GSLS could recover antibody and cellular responses to an inactive bivalent vaccine for Newcastle disease and avian influenza in chickens immunosuppressed by cyclophosphamide [6]. Similar results were also observed in Li’s study [7], in which oral administration of GSLS before immunization of mice with a foot-and-mouth disease vaccine improved specific serum immunoglobulin G (IgG) antibody levels and proliferative splenocyte responses, and significantly increased the numbers of iIELs in the duodenal mucosa and IgA-secreting cells in the lamina propria. These activities were associated with the chemical structures of saponins, in which the carbohydrate moieties might interact with the receptors on the surface of antigen-presenting cells (APCs), promoting antigen phagocytosis and cytokine secretion, while acyl chain domains assist in the delivery of exogenous antigens into APCs and facilitated adaptive immunity [8]. However, the major limitation of using GSLS to improve immunity is their low oral bioavailability and rapid clearance in the gastrointestinal tract, especially for the components with poor water solubility [9]. Therefore, GSLS need to be biochemically stabilized to avoid degradation and enhance absorption in the gastrointestinal tract to achieve maximal beneficial effects.

In recent years, nanoparticle (NP) adjuvant delivery systems have offered attractive features that dictate and direct an immune response, including protection of associated drugs from degradation, increased drug bioavailability, and facilitation of antigen uptake and processing by APCs [10,11,12]. Poly lactic-co-glycolic acid (PLGA), a biodegradable and nontoxic copolymer approved by the United States Food and Drug Administration (FDA) for clinical use [13], has been extensively used to prepare NPs for drug and vaccine delivery. Here, we utilized PLGA to form a nanocarrier for GSLS by the double emulsion/solvent evaporation method. The immunological effects of GSLS nanoparticles (GSLS-NPs) on the phagocytosis and activation of DC2.4 cells were preliminarily investigated in vitro. Then, GSLS-NPs were orally administered to mice before PEDV vaccination, and the adjuvant effects on systemic and intestinal PEDV-specific IgG and IgA antibody responses, neutralizing antibody titers, dendritic cell (DC) activation, and effector and memory T-cell induction in the spleen and mesenteric lymph nodes (MLNs) were further investigated and compared with aqueous solution of GSLS. Moreover, the biosafety of GSLS-NPs was also evaluated in vitro and in vivo. Our results demonstrate that GSLS-NPs may be a potent mucosal adjuvant for the efficient induction of robust and long-term sustained adaptive immune responses against PEDV.

## 2. Material and Methods

### 2.1. Reagents

Standardized GSLS were purchased from Hongjiu Ginseng Industry Co. Ltd. (Jilin, China) and mainly contained Re (29.34%), Rg1 (12.94%), Rd (12.23%), Rb2 (12.22%) and Rc (7.36%), according to HPLC analysis. PLGA (DG-50DLG025, copolymer composition of 50:50, 15–24 kDa, ester terminated) was purchased from Jinan Daigang Biomaterial Co., Ltd. (Jinan, China). All solutions were passed through a gel endotoxin-removing column (Pierce, Shanghai, China), and the endotoxin level of solutions was below 0.5 EU/mL.

### 2.2. Cells and Virus Stock

Vero cells were obtained from the National Animal Gene Research Center of China Agricultural University. Dulbecco’s modified Eagle’s medium (DMEM), RPMI-1640 medium, trypsin (0.25%), fetal bovine serum (FBS), penicillin and streptomycin were purchased from Gibco (Shanghai, China). Cells were maintained in DMEM supplemented with 10% (*v*/*v*) heat-inactivated FBS and antibiotics (100 U/mL penicillin and 8 µg/mL streptomycin) in a humidified 5% CO_2_ incubator at 37 °C.

The murine DC line DC2.4 was obtained from the cell bank at the Chinese Academy of Science (Shanghai, China). DC2.4 cells were cultured in complete RPMI-1640 medium supplemented with 10% (*v*/*v*) FBS and antibiotics (100 U/mL penicillin and 8 µg/mL streptomycin) in a 5% CO_2_ incubator at 37 °C.

The PEDV strain ZJ-ZX2018-C10 (GenBank Accession No. MK250953) was isolated in our laboratory. The virus (1 × 10^6.8^ TCID_50_/mL) was inactivated with 0.05% β-propiolactone (Solarbio, Beijing, China) and used as an inactivated PEDV vaccine. An attenuated PEDV vaccine was obtained by propagating the virus on Vero cells for more than 130 passages (1 × 10^5^ TCID_50_/mL).

### 2.3. Preparation and Characterization of Nanoparticles

GSLS-NPs were prepared by the double emulsion/solvent evaporation method [14]. Briefly, 10 mg of GSLS was dissolved in 1 mL of phosphate-buffered saline (PBS) and emulsified with 10 mL of 20 mg/mL PLGA dichloromethane solution under sonication in an ice bath at 190 W for 2 min. Subsequently, the primary emulsion was diluted with 110 mL of 2% (*w*/*v*) PVA aqueous solution and sonicated on ice at 190 W for 5 min, followed by evaporation of the organic solvents under magnetic stirring overnight. GSLS-NPs were collected from the supernatant colloidal suspension by centrifugation at 3000 rpm for 30 min and lyophilized in a freeze dryer (SCIENTZ-18N, SCIENTZ, Ningbo, China) for further use. Empty PLGA nanoparticles were prepared using a similar method. NP size and the polydispersity index (PDI) were determined with a laser particle size analyzer (Litesizer^TM^ 500, Anton Paar, Graz, Austria). Morphological characterization of NPs was performed using transmission electron microscopy (JEM-1200EX, JEOL, Tokyo, Japan).

A total of 100 mg of freeze-dried NPs was resuspended in 1 mL of acetonitrile to disrupt encapsulation and then diluted in 3 mL of methanol to extract the drugs. The obtained solution was subsequently filtered and subjected to high-performance liquid chromatography (HPLC, Ultra Violet detector, Shimadzu, Kyoto, Japan) with a C18 reversed-phase column (Dikma, Foothill Ranch, CA, USA) at a wavelength of 203 nm. The mobile phase was a mixture of acetonitrile and water (81:19, *v*/*v*) pumped at a rate of 1 mL/min. The HPLC system was controlled using LCSolution software. The encapsulation efficiency (EE) was calculated according to Equation (1):(1)$$\mathrm{EE}\;\left(\%\right)=\frac{\mathrm{weight}\;\mathrm{of}\;\mathrm{the}\;\mathrm{drug}\;\mathrm{in}\;\mathrm{nanoparticles}}{\mathrm{weight}\;\mathrm{of}\;\mathrm{the}\;\mathrm{total}\;\mathrm{drug}}\;\times \;100\%\;$$

### 2.4. Cytotoxicity Assay

The cytotoxicity of GSLS-NPs to DC2.4 cells was tested using a CCK-8 assay. Briefly, DC2.4 cells were plated in a 96-well plate at a density of 1 × 10^5^ cells/mL and incubated overnight to allow cell adhesion. The cells were then treated with GSLS-NPs, GSLS aqueous solution (G-Solution) or empty NPs at different concentrations ranging from 50 to 400 μg/mL. Treatment with LPS (Sigma, Shanghai, China) was used as a positive control. PBS was used to replace the drugs in the negative control (NC) group. After 48 h of incubation, 10 µL of CCK-8 (Dojindo Laboratories, Shanghai, China) was added to each well and incubated for 4 h at 37 °C. The absorbance at 450 nm was evaluated with a microplate reader (Thermo Multiskan MK3, Thermo Fisher Scientific, Shanghai, China). Cell viability was calculated using Equation (2):(2)$$\mathrm{Cell}\;\mathrm{viability}\;\left(\%\right)=\frac{\mathrm{OD}\;\mathrm{value}\;\mathrm{of}\;\mathrm{treated}\;\mathrm{cultures}}{\mathrm{OD}\;\mathrm{value}\;\mathrm{of}\;\mathrm{non}-\mathrm{treated}\;\mathrm{cultures}}\;\times \;100\%\;$$

### 2.5. Cellular Uptake

DC2.4 cells were seeded in a 6-well plate at a density of 1 × 10^5^ cells/mL and then incubated in complete RPMI-1640 medium overnight. After pre-treatment with GSLS-NPs, G-Solution or empty NPs at a final concentration of 100 μg/mL for 4 h, the cells were incubated with the inactivated PEDV antigen for 24 h. Untreated cells served as the NC. After removing the supernatant, the cells were washed three times with PBS and fixed with Immunol Staining Fix Solution (P0098, Beyotime, Shanghai, China). A monoclonal antibody (1:100) against the PEDV nucleocapsid protein was then added and incubated for 1.5 h at 37 °C. DyLight 488-conjugated goat anti-mouse IgG (E032210-01, EARTHOX, San Francisco, CA, USA) (1:200) was used as the secondary antibody. LysoTracker Red (C1046, Beyotime, Shanghai, China) and DAPI (C1005, Beyotime, Shanghai, China) were used to stain the lysosomes and nuclei, respectively. The cells were observed under an inverse confocal laser scanning microscope (FV1000, Olympus, Tokyo, Japan). The images were acquired using Olympus confocal software (FV10-ASW 3.1). In addition, a monoclonal antibody against the PEDV nucleocapsid protein was labeled with the fluorescent dye APC by using an APC conjugation kit (ab201807, Abcam, Shanghai, China). The relative fluorescence intensity of intracellular PEDV was quantified on a FACSCanto^TM^ (BD Biosciences, San Diego, CA, USA).

### 2.6. Animal Studies

Female BALB/c mice aged 6–8 weeks were purchased from the Shanghai Laboratory Animal Center (Shanghai, China). Mice were housed under specific pathogen-free conditions at a stable temperature (24 ± 1 °C) and randomly divided into 10 groups, with 6 mice in each group. All experiments related to animal care and use were approved by the Animal Care Committee of the Zhejiang Academy of Agricultural Sciences in accordance with the recommendations of the National Institutes of Health Guide for the Care and Use of Laboratory Animals (Ethics protocol No. 2020001).

In one experiment, mice were gavaged daily for 7 days according to their group: GSLS-NPs 1 (2.5 mg/kg), GSLS-NPs 2 (5 mg/kg), GSLS-NPs 3 (10 mg/kg), G-Solution 1 (2.5 mg/kg), G-Solution 2 (5 mg/kg), G-Solution 3 (10 mg/kg), empty NPs (10 mg/kg), LPS (50 µg/kg) or saline. Four hours after the final gavage, the mice received 1 × 10^4^ TCID_50_ of attenuated PEDV vaccine via i.m. injection (day 0), and a booster injection of the inactivated vaccine (1 × 10^5.8^ TCID_50_) was administered on day 21. The mice treated with saline alone served as the NC. On day 35, blood and feces were collected for antibody determination. The blood was centrifuged at 1000× *g* for 10 min to separate the serum. Three to four fresh fecal pellets were collected into a micro-tube and weighed, and a 10× volume (per gram of feces) of extraction buffer (5% FBS and 0.02% sodium azide in PBS) was added to each tube to normalize the variation in the fecal weight among different mice. The fecal samples were vortexed vigorously and then centrifuged at 10,000× *g* for 10 min. The supernatant was collected for further use.

In the other experiment, mice were gavaged daily with GSLS-NPs, G-Solution or empty NPs at a dose of 5 mg/kg for 7 days. Four hours after the final gavage, the mice were immunized with 1 × 10^4^ TCID_50_ of attenuated PEDV vaccine via the i.m. route (day 0); then, they received a booster immunization with 1 × 10^5.8^ TCID_50_ of inactivated vaccine on day 21. Blood was harvested on days 2 and 23 for biochemical analysis and collected on day 35 for antibody determination. Feces and small intestines were collected on day 35 for immunological testing. Three-centimeter sections of the duodenum, jejunum and ileum were extracted for immunohistochemical and hematoxylin-eosin (H.E.) staining. In addition, intestinal samples were cut into pieces and vortexed with 400 μL of extraction buffer. The supernatant was harvested after centrifugation for antibody detection. Spleens and MLNs were harvested on day 23 for analysis of DC activation and collected on day 35 for evaluation of cytokine responses and T-cell activities. The spleens and MLNs were homogenized and passed through a steel mesh to obtain a cell suspension. After centrifugation at 380× *g* for 10 min, the cell pellets were washed in Hank’s balanced salt solution and re-suspended in complete RPMI-1640 medium for further use.

### 2.7. Enzyme-Linked Immunosorbent Assay (ELISA) for Antibody Determination

Blood collected on day 35 was centrifuged at 1000× *g* for 10 min to separate the serum. PEDV-specific IgA, IgG, IgG1 and IgG2a antibodies were measured by ELISA. Briefly, polyvinyl 96-well microplates pre-coated with the PEDV antigen were obtained from PEDV antibody detection kits (HomSun Bio-Technology, Hangzhou, China). After adding either 100 μL of diluted serum samples (1:100) or fecal fluids (1:2), the plates were incubated at 37 °C for 1 h. After washing, 100 μL of horseradish peroxidase (HRP)-conjugated goat anti-mouse IgA (1:2000), IgG (1:5000), IgG1 (1:2000) or IgG2a (1:2000) was added, and the plate was incubated at 37 °C for 1 h. After another round of washing, 100 μL of 3,3′,5,5′-tetramethyl benzidine substrate solution was added to each well, and the reaction was stopped using a stop solution after 15 min. The optical density (OD) of the wells was read at 450 nm by a microplate reader (Thermo Multiskan MK3, Thermo Fisher Scientific, Shanghai, China).

### 2.8. Immunohistochemistry and Haematoxylin-Eosin Staining

IgA^+^ cells were identified with an immunohistochemical staining method described previously [15]. Briefly, fixed intestinal samples were embedded in paraffin, sectioned at a thickness of 6 µm, and mounted onto glass slides. Endogenous peroxidase activity was blocked by incubation with 3% H_2_O_2_ in methanol for 20 min. After rinsing three times with PBS, the slides were successively incubated with a goat anti-mouse IgA antibody (1:400) and HRP-conjugated rabbit anti-goat IgG antibody (1:250). 3, 3′-Diaminobenzidine (DAB) (BOSTER, Wuhan, China) detection was performed according to the manufacturer’s instructions. Images were captured with a DS-U3 camera interface (Nikon, Tokyo, Japan) and evaluated using the image analysis software Image-Pro Plus 6.0. Three fields per slide were randomly selected at 400× magnification, and the integrated optical density (IOD) was analyzed.

After hematoxylin–eosin staining, the number of iIELs in five different fields of intestinal villi on each slide was counted to statistically analyze the data.

### 2.9. Neutralization Assay

Serum and fecal fluids were inactivated at 56 °C for 30 min and serially diluted 1:2. The serial dilutions of the samples were mixed with equal volumes of 100 TCID_50_ of PEDV virus and incubated at 37 °C for 1 h. The mixtures were then transferred to Vero cell monolayers on 96-well plates. The plates were incubated for 7 days at 37 °C in a 5% CO_2_ atmosphere, and the cytopathic effect (CPE) was recorded. Neutralizing antibody titer that prevented CPE in 50% of the wells was calculated by the Reed and Muench method [16].

### 2.10. Cytokine Release

DC2.4 cells were plated at a density of 1 × 10^5^ cells/mL in a 24-well plate and treated with GSLS-NPs, G-Solution or empty NPs at a final concentration of 100 μg/mL for 72 h. LPS- and PBS-treated cells served as the positive control and NC, respectively. The supernatant was collected to determine the concentrations of IL-1β, TNF-α and IL-6. Mononuclear cells from the spleen or MLNs of mice were re-stimulated with the inactivated PEDV antigen (10 μg/mL) for 72 h at 37 °C in a 5% CO_2_ atmosphere. The supernatant was harvested to detect the levels of IFN-γ, IL-10 and IL-6. Cytokine levels were measured with commercial ELISA kits (Neobioscience, Shanghai, China) according to the manufacturer’s instructions.

### 2.11. Flow Cytometry

To detect the activation of DCs, mononuclear cells from the spleen or MLNs were stained with PE/Cy7-conjugated anti-CD11c (clone N418), PE-conjugated anti-MHC II (clone M5/114.15.2), APC-conjugated anti-PEDV, FITC-conjugated anti-CD11b (clone M1/70), PerCP-Cy5.5-conjugated anti-CD8α (clone 53-6.7) and PerCP-Cy5.5-conjugated anti-CD103 (clone 2E7). To evaluate memory T-cell responses, mononuclear cells from the spleen or MLNs were stained with FITC-conjugated anti-CD3e (clone 145-2C11), APC-conjugated anti-CD4 (clone GK1.5), PerCP-Cy5.5-conjugated anti-CD8a (clone 53-6.7), PE-Cy7-conjugated anti-CD44 (clone IM7) and PE-conjugated anti-CD62L (clone MEL-14). The PE/Cy7-conjugated anti-CD11c and PerCP-Cy5.5-conjugated anti-CD103 antibodies were purchased from BioLegend (San Diego, CA, USA). The APC-conjugated anti-PEDV antibody was prepared in our laboratory. The PE-conjugated anti-CD62L antibody was purchased from Southern Biotech (Birmingham, AL, USA). The other antibodies were obtained from MultiSciences (Hangzhou, China). All data were collected on a FACSCanto^TM^ (BD Biosciences, San Diego, CA, USA) and analyzed using FlowJo (Version 10.0).

### 2.12. Blood Biochemistry Analysis

Serum was assayed for the levels of alanine aminotransferase (ALT), aspartate aminotransferase (AST), lactate dehydrogenase (LDH), alkaline phosphatase (ALP), total bilirubin (TBIL) and blood urea nitrogen (BUN) with appropriate assay kits (Nanjing Jiancheng Bioengineering Institute, Nanjing, China) following the manufacturer’s instructions.

### 2.13. Statistical Analysis

Data are expressed as the mean ± standard deviation (SD). All statistical analyses were performed with GraphPad Prism 7.0 software (San Diego, CA, USA); an unpaired Student’s *t* test was used for two groups and one-way ANOVA with Tukey’s multiple comparisons test was used for three or more groups. A *p* value < 0.05 was considered statistically significant. All experiments were conducted a minimum of three times.

## 3. Results

### 3.1. Synthesis and Characterization of Nanoparticles

As shown in Figure 1A, GSLS-NPs and empty NPs exhibited a smooth surface and spherical morphology. The center of empty NPs was bright, while that of GSLS-NPs showed a dark area, which suggested that GSLS was successfully loaded into the PLGA shell. The average diameter of GSLS-NPs was 230.7 ± 5.3 nm, with a PDI of 0.1348 ± 0.0276 (Figure 1B,C). There were no significant differences in particle size or PDI between GSLS-NPs and empty NPs. As determined by HPLC analysis, the encapsulation efficiencies of GSLS-NPs were 44.32 ± 6.93% for Re, 51.20 ± 6.25% for Rg1, 69.34 ± 7.94% for Rd, 36.63 ± 4.36% for Rb2, and 58.56 ± 4.36% for Rc (Figure 1D).

### 3.2. Viability and Activation of DCs Induced by GSLS-NPs In Vitro

DCs are highly specialized in processing and presenting antigens to activate lymphocytes. An in vitro cytotoxicity experiment was conducted with the murine DC line DC2.4 using a CCK-8 assay. Cells were cultured with GSLS-NPs at concentrations of 50–400 µg/mL, and no significant cell death was observed over 48 h (Figure 2A). To investigate the potential of GSLS-NPs to enhance the phagocytic capacity of DCs in vitro, the uptake of a PEDV antigen by DC2.4 cells was first evaluated by laser confocal microscopy detection. As shown in Figure 2B, GSLS-NPs effectively induced antigen internalization by DC2.4 cells, and the fluorescence intensity of the intracellular antigen achieved with GSLS-NPs was obviously stronger than that achieved by treatment with G-Solution, empty PLGA NPs or PBS. G-Solution had a slight enhancing effect on antigen internalization when compared with empty PLGA NPs or PBS. A negligible effect was found for empty PLGA NPs. These results were further confirmed by flow cytometric analysis. As shown in Figure 2C, when compared with empty NPs or PBS, GSLS-NPs (*p* < 0.001) and G-Solution (*p* < 0.05) significantly promoted the antigen uptake ratio of DC2.4 cells. Surprisingly, from the results of the flow cytometric analysis, no significant differences in the antigen uptake ratio were observed between the GSLS-NP and G-Solution treatments.

Next, we analyzed the pro-inflammatory cytokines expressed by DC2.4 cells in response to stimulation with GSLS-NPs, G-Solution or empty NPs. As shown in Figure 2D, DC2.4 cells treated with GSLS-NPs secreted significantly higher levels of IL-1β, TNF-α and IL-6 than untreated cells (*p* < 0.0001) or cells treated with G-Solution or empty NPs (*p* < 0.0001 for IL-1β and IL-6, *p* < 0.001 for TNF-α). Interestingly, the production of IL-1β and IL-6 in the GSLS-NP group was even higher than that in the lipopolysaccharide (LPS)-treated group (*p* < 0.0001 for IL-1β, *p* < 0.01 for IL-6). However, little effect was found for treatment with G-Solution or empty NPs.

### 3.3. Oral Administration of GSLS-NPs Enhances PEDV-Specific Antibody Responses in the Serum and Gastrointestinal Tract

To evaluate the adjuvant effect of GSLS-NPs on PEDV vaccination, PEDV-specific antibody responses in mice were analyzed 14 days after a booster immunization. When compared with the vaccine-only control, the addition of the adjuvant GSLS-NPs at a dose of 5 or 10 mg/kg significantly enhanced the IgG responses in the serum (*p* < 0.05), but only the dose of 5 mg/kg could strongly increase the titer of specific IgA antibodies in the feces (*p* < 0.05) (Figure 3A). G-Solution or empty NPs showed little enhancing effect on the vaccine-induced serum IgG and fecal IgA antibody responses. As a positive control, LPS was effective only in elevating serum IgG responses (*p* < 0.05), not in increasing fecal IgA antibody levels.

As shown in Figure 3B, the prominent IgG isotypes elevated in the serum by vaccination were IgG1 and IgG2a, which were significantly higher in the adjuvant group treated with GSLS-NPs than in the vaccine-only control group (*p* < 0.001 for IgG1 and IgG2a) and vaccine-plus-empty-NPs group (*p* < 0.001 for IgG1, *p* < 0.01 for IgG2a). Mice immunized with the vaccine plus G-Solution generated similar levels of serum IgG1 but lower levels of IgG2a (*p* < 0.05) than mice immunized with the vaccine supplemented with GSLS-NPs. The serum IgG1 isotype titer induced by addition of G-Solution was apparently higher than that in the vaccine-only control (*p* < 0.01) and the addition of empty NPs (*p* < 0.05). Furthermore, the IgA-secreting cells in each region of the small intestine were analyzed using immunohistochemistry. As shown in Figure 3C,D, the addition of GSLS-NPs significantly increased the numbers of IgA-secreting cells in the duodenum, jejunum and ileum when compared with the vaccine-only control, vaccine plus G-Solution or vaccine plus empty NPs (*p* < 0.0001). The number of IgA-secreting cells in the whole small intestine was similar among mice receiving the vaccine alone or vaccine plus G-Solution or empty NPs. Notably, vaccination alone increased only IgA-secreting cells in the duodenum when compared to no immunization (*p* < 0.05). As expected, the GSLS-NPs adjuvant also elevated PEDV-specific IgA antibody levels in segments of the duodenum (*p* < 0.0001), jejunum (*p* < 0.01) and ileum (*p* < 0.05) compared with the vaccine alone (Figure 3E), which was consistent with the increased populations of IgA-secreting cells. However, compared to the addition group of G-Solution or empty NPs, GSLS-NPs elicited stronger IgA antibody responses in only the duodenum (*p* < 0.0001) and jejunum (*p* < 0.01), not in the ileum, which was different from the IgA-secreting cell population results.

The neutralizing antibody titers in serum and feces were further analyzed. As shown in Figure 3F, mice receiving the vaccines alone generated a higher serum neutralizing antibody titer than unimmunized mice (*p* < 0.0001), but no difference was observed in the feces. GSLS-NPs significantly promoted neutralizing antibody responses in both the serum and the feces when compared with the vaccine-only control (*p* < 0.0001), vaccine plus G-Solution (*p* < 0.001) or empty NPs (*p* < 0.0001). However, the G-Solution and empty NPs had little effect on the neutralizing antibody response in either the serum or the feces compared to the vaccine alone.

### 3.4. Oral Administration of GSLS-NPs Increases iIEL Proliferation

iIELs are small round cells with densely stained nuclei located within the epithelial cell layer in the small intestine. As shown in Figure 4, intramuscular (i.m.) inoculation of PEDV vaccines had little enhancing effect on the number of iIELs. Mice receiving the GSLS-NPs adjuvant generated more iIELs in the intestinal mucosa than mice immunized with the vaccine alone (*p* < 0.01), vaccine plus G-Solution (*p* < 0.05) or vaccine plus empty NPs (*p* < 0.01). However, no significant increases in iIELs were observed in the G-Solution or empty-NPs addition group when compared with the vaccine-only control group.

### 3.5. GSLS-NPs Regulate Distinct Subsets of Splenic and Intestinal DCs

DCs are critical initiators and modulators of adaptive immune responses, and as such, targeting DCs directly is an approach to enhance responses to vaccines. In this study, approximately 40% of mature CD11c^+^ DCs from the spleen (Figure 5A,B) and MLNs (Figure 6A,B) highly expressed major histocompatibility complex (MHC) class II and acquired the PEDV antigen after immunization with the vaccine plus GSLS-NPs, which was significantly higher than the rates in the vaccine-only control (*p* < 0.01 in the spleen and MLNs) and empty-NPs-addition (*p* < 0.05 in the spleen, *p* < 0.01 in the MLNs) groups. Moreover, the population of PEDV-positive (PEDV^+^) DCs in the MLNs of mice receiving G-Solution was significantly lower than that in mice treated with GSLS-NPs (*p* < 0.05), which was similar to those in the vaccine-only control and vaccine-plus-empty-NPs groups. Notably, i.m. inoculation of PEDV vaccines enhanced the antigen uptake by DCs only in the spleen (*p* < 0.05).

Multiple subsets of DCs with distinct functions and roles in the priming of T-cell responses exist [14]. Thus, we analysed various subsets of MHCII^+^ CD11c^+^ DCs to determine which specific subsets were involved in the uptake of the PEDV antigen. As shown in Figure 5C,D, the CD11b^+^ CD8α^−^ subset was the most frequent PEDV^+^ DC population in the spleen of immunized mice, and mice receiving the vaccine plus GSLS-NPs showed an apparently higher proportion of CD11b^+^ CD8α^−^ DCs than mice immunized with the vaccine alone (*p* < 0.01), vaccine plus G-Solution (*p* < 0.05) or vaccine plus empty NPs (*p* < 0.01). Furthermore, the PEDV antigen was mainly associated with CD11b^−^ CD103^+^ DCs in the MLNs of mice receiving GSLS-NPs adjuvant (Figure 6C,D), and this population was significantly larger in these mice than in those treated with the vaccine alone (*p* < 0.01), vaccine supplemented with G-Solution (*p* < 0.05) or empty NPs (*p* < 0.01), suggesting that GSLS-NPs may have a more profound impact on migratory CD11b^−^ CD103^+^ DCs than on other cells. However, negligible differences in the intestinal DC subsets were found among the vaccine-only, vaccine-plus-G-Solution, vaccine-plus-empty-NPs and non-immunized groups.

### 3.6. GSLS-NPs Expand Memory and Effector T-Cell Responses

T lymphocyte (CD3^+^) subpopulations in the spleen and MLNs were analyzed 14 days after a secondary immunization. As shown in Figure 7A,B, the GSLS-NPs adjuvant was more effective than G-Solution and empty NPs in triggering T-cell differentiation when delivered orally. The proportion of the CD4^+^ subpopulation was significantly higher in spleens and MLNs from mice immunized with the vaccine supplemented with GSLS-NPs than in those from mice immunized with the vaccine alone (*p* < 0.001 in the spleen, *p* < 0.0001 in the MLNs), vaccine plus G-Solution (*p* < 0.05 in the spleen and MLNs) or vaccine plus empty NPs (*p* < 0.01 in the spleen, *p* < 0.0001 in the MLNs). Increased CD8^+^ T cells were observed only in the MLNs, not the spleen, in the GSLS-NPs adjuvant group compared to other immunized groups (*p* < 0.05). Notably, parenteral inoculation increased only CD4^+^ (*p* < 0.001) cells, not CD8^+^ T cells, in the spleen but had little impact on the T-cell subpopulation in the MLNs.

Effective activation of memory T cells contributes to long-lasting immunity, which helps to strengthen adaptive immune responses [17]. As shown in Figure 7C, GSLS-NPs increased specific memory (CD44^hi^ CD62L^hi^) and effector (CD44^hi^ CD62L^lo^) CD4^+^ T cells in both the spleen (*p* < 0.0001 for memory, *p* < 0.01 for effector) and the MLNs (*p* < 0.0001 for memory, *p* < 0.01 for effector) compared to the vaccine-only control. Moreover, the proportions of memory and effector CD4^+^ T cells in the spleen (*p* < 0.0001 for memory, *p* < 0.01 for effector) and the MLNs (*p* < 0.0001 for memory, p < 0.05 for effector) were higher in the GSLS-NPs adjuvant group than in the empty NPs addition group. When compared with treatment of the G-Solution, the addition of GSLS-NPs showed elevated numbers of memory CD4^+^ T cells in both the spleen and MLNs (*p* < 0.0001), while a larger effector CD4^+^ T-cell subpopulation was observed only in the spleen (*p* < 0.01), not in the MLNs. Notably, parenteral inoculation increased only effector CD4^+^ T cells in the spleen, not in the MLNs (*p* < 0.05), and had little enhancing effect on memory CD4^+^ T cells in either the spleen or the MLNs. Furthermore, the results for specific memory (CD44^hi^ CD62L^hi^) and effector (CD44^hi^ CD62L^lo^) CD8^+^ T cells are presented in Figure 7D. In the spleen, rare significant differences were found among all the immunized groups in either memory or effector CD8^+^ T cells. In the MLNs, mice immunized with the vaccine plus GSLS-NPs exhibited significantly higher percentages of memory and effector CD8^+^ T cells than mice receiving the vaccine alone (*p* < 0.0001 for memory, *p* < 0.001 for effector) or vaccine plus G-Solution or empty NPs (*p* < 0.0001 for memory, *p* < 0.05 for effector).

We further evaluated mononuclear cells isolated from the spleen or MLNs of immunized mice for cytokine production upon PEDV antigen re-stimulation in vitro. As shown in Figure 7E, the vaccine alone increased the level of IL-6 only in the spleen compared to no immunization (*p* < 0.01). Oral administration of GSLS-NPs significantly increased the levels of Th1 (IFN-γ), Th2 (IL-10) and Th17 (IL-6) cytokines in both the spleen (*p* < 0.01 for IFN-γ, *p* < 0.0001 for IL-10 and *p* < 0.001 for IL-6) and the MLNs (*p* < 0.05 for IFN-γ, *p* < 0.0001 for IL-10 and IL-6) compared to vaccine-only immunization. When compared with the administration of empty NPs, treatment with GSLS-NPs remarkably elevated the production of IFN-γ, IL-10 and IL-6 in the spleen (*p* < 0.01 for IFN-γ, *p* < 0.0001 for IL-10 and *p* < 0.01 for IL-6) and MLNs (*p* < 0.05 for IFN-γ, *p* < 0.0001 for IL-10 and IL-6). Moreover, compared with treatment of the G-Solution, the addition of GSLS-NPs exhibited higher levels of IL-10 and IL-6 in both the spleen (*p* < 0.0001 for IL-10 and *p* < 0.01 for IL-6) and the MLNs (*p* < 0.0001 for IL-10 and IL-6), while a higher IFN-γ level was observed only in the spleen (*p* < 0.01), not in the MLNs. Notably, oral administration of the G-Solution increased the production of Th2-type cytokines (IL-10) in both the spleen and the MLNs compared to treatment with empty NPs or the vaccine alone (*p* < 0.001).

### 3.7. Biosafety of GSLS-NPs as an Oral Adjuvant

To evaluate the safety of oral delivery of GSLS-NPs, blood biochemistry analysis was conducted 2 days after the primary and secondary immunizations. As shown in Figure 8, no significant treatment-related differences in the serum levels of ALT, AST, LDH, ALP, TBIL or BUN were recorded among all the groups, indicating no overall adverse effects on hepatic or renal function.

## 4. Discussion

To date, parenteral vaccination to prevent PEDV infection has achieved little success, as the current vaccines fail to induce sufficient mucosal immunity, which plays a crucial role in immune defenses against PEDV infection. One attractive way to circumvent this issue would be the use of an effective mucosal adjuvant to sensitize the immune status of the intestinal mucosa before vaccination. In this study, we prepared a nano-based delivery system for GSLS and explored its mucosal adjuvant potential in the context of PEDV vaccine.

The GSLS-NPs exhibited a spherical shape with an average diameter of approximately 230 nm and a low PDI value, suggesting the narrow size distribution of the NPs. According to the HPLC analysis results, the encapsulation efficiency was high for the hydrophilic saponins of GSLS, including Re, Rg1, Rd and Rc, but low for Rb2, which possesses lipophilic properties, indicating that the water-in-oil-in-water (w/o/w) method is more suitable to encapsulate water-soluble agents, as previously reported [18]. Based on the results of the in vitro viability assay and in vivo blood biochemistry analysis, no apparent cytotoxicity or hepatic or renal dysfunction was observed after treatment with GSLS-NPs, indicating that GSLS-NPs have a good biosafety profile.

DCs, as one of the most important professional APC populations, have a specific function in antigen presentation, which is distinct from that of macrophages and B cells. DCs can present extracellular and intracellular antigens by direct presentation and cross-presentation, which promote the differentiation and activation of CD4^+^ and CD8^+^ T cells, respectively [19]. In this work, we initially found that GSLS-NPs significantly enhanced antigen internalization and cytokine responses by DC2.4 cells, suggesting that GSLS-NPs could promote the activation of DCs to enhance immune activities, which was further proven in animals. Distinct DC subsets may be programmed for different functions depending on the nature of the stimulus [20]. Oral administration of GSLS-NPs tended to promote the differentiation of splenic DCs into the CD11b^+^ CD8α^−^ phenotype, which is the population specifically responsible for antigen presentation in the spleen. CD11b^+^ CD8α^−^ DCs can further present antigens exclusively to CD4^+^ T cells and prime antigen-specific immune responses [19], which can be seen from the biased increase in CD4^+^ T cells and balanced improvements in Th1, Th2 and Th17 cytokine responses in spleen from the mice immunized with the vaccine plus GSLS-NPs compared to those from mice in other immunized groups. Notably, splenic CD4^+^ T cells from mice immunized with the vaccine supplemented with GSLS-NPs further differentiated into specific memory and effector CD4^+^ T-cell subsets, both of which were more abundant in this group than in the other immunized groups. Memory T cells form the cellular basis for accelerated protection upon re-exposure to the same pathogen, which is a hallmark of adaptive immunity [21,22]. In contrast, splenic CD11b^+^ CD8α^−^ DCs activated by parenteral inoculation alone preferentially primed effector CD4^+^ T cells but not memory CD4^+^ T cells and mainly promoted the Th17 immune response. It was surprising to note that none of the treatment groups showed an effect on CD8α^+^ DCs, resulting in little activation of CD8+ T cells in the spleen. In contrast, in the draining MLNs, GSLS-NPs guided the recruitment of intestinal CD11b^−^ CD103^+^ DCs, which can cross-present antigens to CD8^+^ T cells and promote the differentiation of CD4^+^ T cells into Th1 and Foxp3+ regulatory T (Treg) cells [23,24]. The activated CD4^+^ and CD8^+^ T cells in the MLNs of mice immunized with the vaccine plus GSLS-NPs further differentiated into memory and effector T-cell subsets and expressed higher levels of Th1, Th2 and Th17 cytokines than those in other immunized groups, indicating that robust cytotoxic T-cell activities and that balanced Th1/Th2/Th17 responses were induced in the draining MLNs by the additional administration of GSLS-NPs. It has been reported that CD11b^−^ CD103^+^ DCs can activate the gut-homing receptors CCR9 and integrin α4β7 on the surface of T cells to promote the homing of effector T cells to the effector site in the intestinal mucosa and protect against pathogens [25]. Correlatively, iIELs, which form one of the first lines of immune defense in gut tissue, were also significantly increased in the intestinal epithelium of mice immunized with the vaccine plus GSLS-NPs compared with that of other immunized mice, suggesting that intestinal defense against PEDV invasion was established. However, parenteral inoculation alone had little promotive effect on cellular immunity in the intestine. Except for the induction of Th2-biased activities, the administration of the mixture containing major saponins of GSLS elicited cellular immune responses in the spleen and MLNs similar to those in the vaccine-only group.

B cells are an essential part of the adaptive immune system, acting as drivers of humoral immune responses through T-cell-dependent and T-cell-independent pathways, and humoral immunity, especially intestinal IgA responses, plays an important role in immune protection against PEDV infection [26]. Our antibody results demonstrated that the ability of GSLS-NPs to improve antibody responses was not dose-dependent, and the optimal GSLS-NPs dose for adjuvant activities appeared to be 5 mg/kg, which elicited a stronger serum IgG antibody response, specifically responses involving the IgG1 and IgG2a isotypes, 14 days after a booster immunization than vaccination alone. The predominant IgG subclass changes progressively during an immune response, which is dependent on the pattern of cytokines secreted by CD4^+^ T cells [15]. In mice, IL-10 favors IgG1 isotype (Th2 type) production by activated B cells, while IFN-γ promotes IgG2a production (Th1 type). The results indicated that GSLS-NPs stimulated unbiased Th1 and Th2 immune responses in mice, consistent with the cytokine responses. Furthermore, the numbers of IgA-secreting plasma cells in the mucosa of the duodenum, jejunum and ileum were increased by the additional administration of GSLS-NPs, a result corresponding to the finding of more abundant IgA antibodies in all segments of the small intestine. Surprisingly, the large number of IgA-secreting cells induced by GSLS-NPs in the ileum did not produce sufficient IgA antibodies, the level of which was higher than only that in the vaccine-only group but similar to those in other immunized mice. As expected, the additional administration of GSLS-NPs compared to the other treatments enhanced the levels of neutralizing antibodies in the serum and intestine, which are critical for protection against PEDV infection, especially those in the intestine. Interestingly, parenteral inoculation alone could induce a PEDV-specific serum IgG response with balanced IgG1 and IgG2a subtypes and an intestinal IgA response limited to the duodenum but had no effect on IgA antibodies in the jejunum and ileum, which are the main invasion and replication sites of PEDV. Moreover, parenteral inoculation induced neutralizing antibodies only in the serum, not in the local mucosa. The aqueous solution of GSLS preferentially enhanced the PEDV-specific IgG1 isotype response in the serum, which correlated with the increased induction of IL-10 in the spleen. However, it was surprising that the aqueous solution had a rare enhancing effect on the serum total IgG antibody response and local IgA antibody response compared to the vaccine alone, which led to a limited effect on serum and intestinal neutralizing antibodies.

## 5. Conclusions

In this study, we successfully developed a safe and efficient well-characterized nanoparticle delivery system for GSLS that offers a promising strategy for improving the efficacy of parenteral vaccines against PEDV. In addition to enhancing systemic cellular and humoral immune responses, oral administration of GSLS-NPs at the dose of 5 mg/kg before parenteral inoculation potently enhanced the phagocytic function and activation of DCs, expanded the activities of effector and memory T cells, and increased PEDV-specific IgA and neutralizing antibodies in the local intestinal mucosa, showing great advantages over the aqueous solution of GSLS. Therefore, GSLS-NPs could be an attractive mucosal adjuvant for parenteral vaccines commonly used in the field to prevent PEDV infection.

## Figures and Tables

**Figure 1 vaccines-10-01810-f001:**
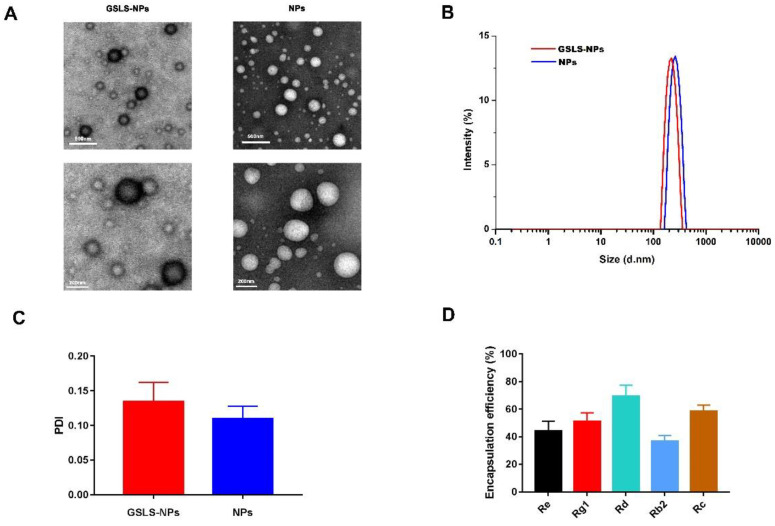
Characterization of nanoparticles. (**A**) Transmission electron microscopy images of GSLS-NPs and empty NPs. Low magnification, scale bar = 500 nm. High magnification, scale bar = 200 nm. (**B**) Particle Size. (**C**) PDI value. (**D**) Encapsulation efficiency.

**Figure 2 vaccines-10-01810-f002:**
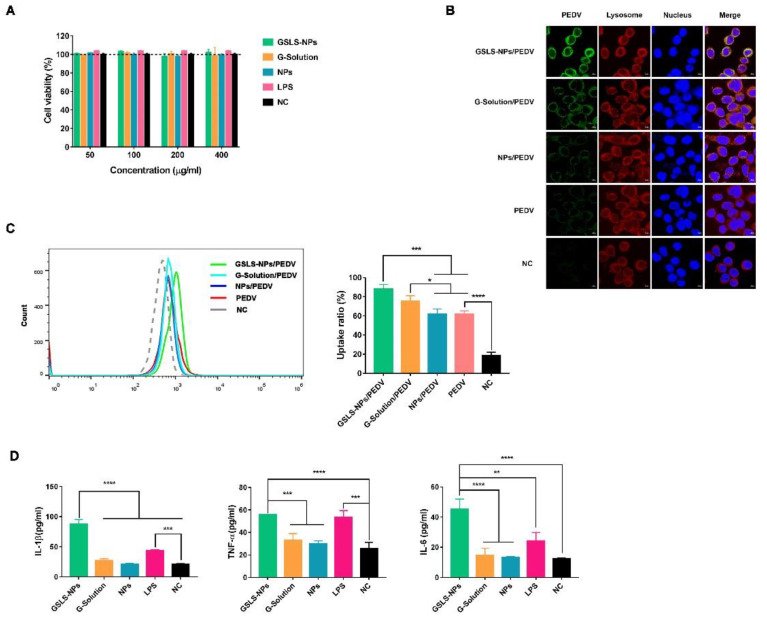
Effect of GSLS-NPs on the activation of DC2.4 cells. (**A**) Viability of DC2.4 cells measured with a CCK-8 assay after incubation with GSLS-NPs, G-Solution or empty NPs for 48 h. LPS was used as a positive control, and untreated cells served as a negative control (NC). (**B**,**C**) Representative fluorescence microscopy images and flow cytometric analysis of antigen uptake by DC2.4 cells after pre-treatment with GSLS-NPs, G-Solution, empty NPs or PBS for 4 h and incubation with the PEDV antigen for the following 24 h. (**D**) Production of cytokines by DC2.4 cells after 72 h of incubation. The experiments were performed in triplicate. Data represent the averages of triplicate samples from three identical experiments and are presented as the mean ± SD. Statistical analysis was performed using an unpaired Student’s *t* test. * *p* < 0.05, ** *p* < 0.01, *** *p* < *0*.001, **** *p* < 0.0001.

**Figure 3 vaccines-10-01810-f003:**
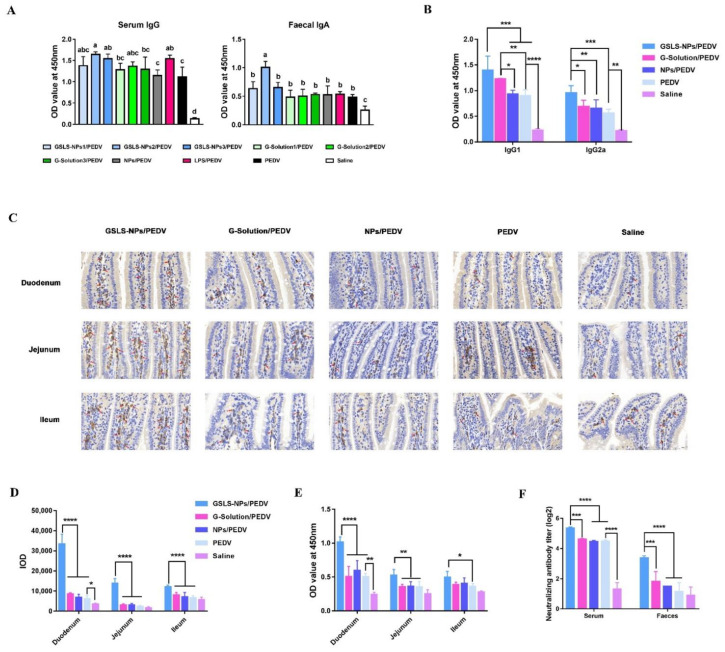
GSLS-NPs enhanced PEDV-specific antibody responses in the blood and intestinal mucosa. (**A**) PEDV-specific serum IgG and fecal IgA antibodies. Mice (n = 6) were gavaged daily with saline, empty NPs (10 mg/kg), G-Solution (2.5–10 mg/kg) or GSLS-NPs (2.5–10 mg/kg) for 7 days. LPS (50 µg/kg) was used as a positive control. After the final gavage, the mice were intramuscularly immunized with an attenuated PEDV vaccine (1 × 10^4^ TCID_50_) on day 0, followed by a booster immunization with an inactivated vaccine (1 × 10^5.8^ TCID_50_) on day 21. Non-immunized mice served as the negative control (NC). Serum and feces were collected on day 35 for antibody analysis by ELISA. Data are expressed as the mean ± SD. Statistical analysis was performed using one-way ANOVA with Tukey’s multiple comparisons test. Different letters represent significant differences (*p* < 0.05). (**B**) Serum IgG isotypes. Mice (n = 6) were gavaged daily with saline, empty NPs (5 mg/kg), G-Solution (5 mg/kg) or GSLS-NPs (5 mg/kg) for 7 days and immunized as described above. The mice were euthanized on day 35. (**C**) IgA-secreting cells in the duodenum, jejunum and ileum. (**D**) Integrated optical density (IOD) of IgA-secreting cells. (**E**) PEDV-specific IgA antibody responses in the duodenum, jejunum and ileum. (**F**) Neutralizing antibody titer. Experiments were performed in triplicate. Data are presented as the mean ± SD. Stars denote significant differences: * *p* < 0.05, ** *p* < 0.01, *** *p* < *0*.001, **** *p* < 0.0001. Unpaired Student’s *t* test.

**Figure 4 vaccines-10-01810-f004:**
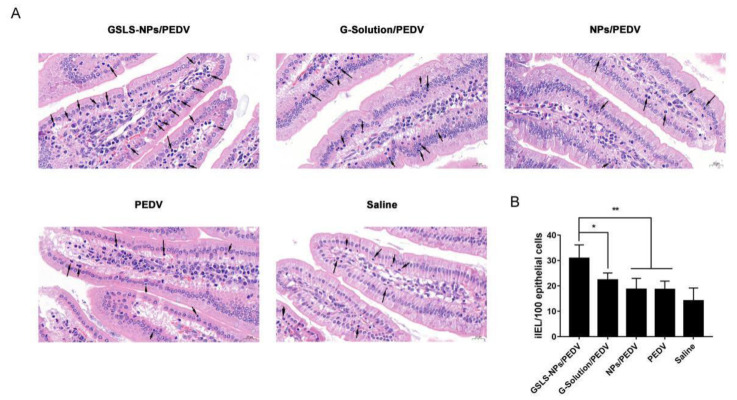
GSLS-NPs increased intestinal intraepithelial lymphocytes (iIELs) in the mucosa of the small intestine. Mice (n = 6) were gavaged daily with saline, empty NPs, G-Solution or GSLS-NPs for 7 days. After the final gavage, the mice were immunized as described above and euthanized on day 35. Non-immunized mice served as the NC. (**A**) Representative images of iIELs in the epithelial cell layer of the small intestine identified by hematoxylin-eosin staining (indicated as arrows). Scale bar = 20 µm. (**B**) iIEL count per 100 epithelial cells in five different fields of intestinal villi on each slide. Data are presented as the mean ± SD. * *p* < 0.05, ** *p* < 0.01. Unpaired Student’s *t* test.

**Figure 5 vaccines-10-01810-f005:**
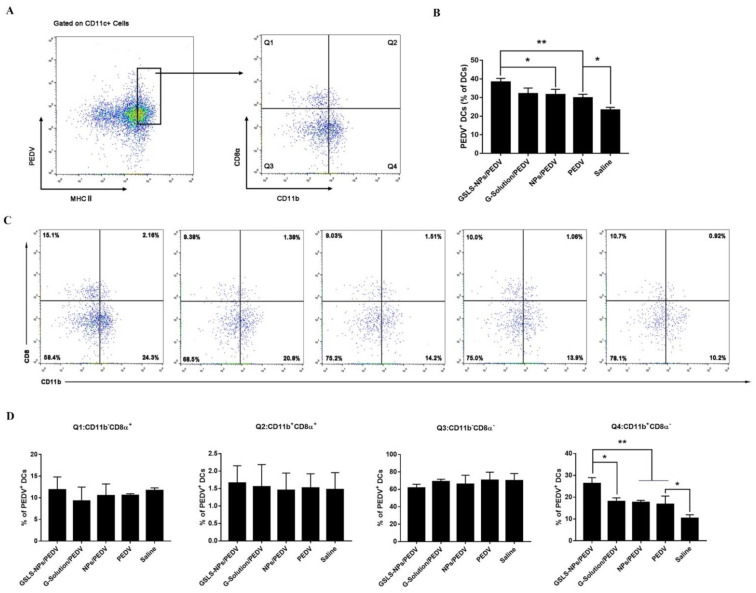
Responses of dendritic cell subsets in the spleen. Mice (n = 6) were gavaged and immunized as described above, and splenocytes were collected after euthanasia on day 23. (**A**,**B**) Frequency of PEDV-positive DCs. (**C**) Representative dot plot of the DC subset distribution. (**D**) Frequencies of PEDV-positive DC subsets. Data are expressed as the mean ± SD. * *p* < 0.05, ** *p* < 0.01. Unpaired Student’s *t* test.

**Figure 6 vaccines-10-01810-f006:**
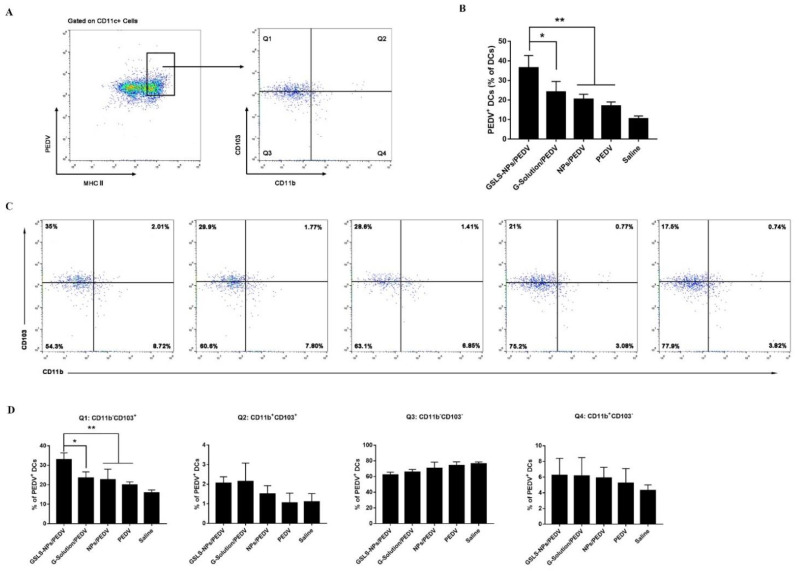
Responses of dendritic cell subsets in the MLNs. Mice (n = 6) were gavaged and immunized as described above, and mononuclear cells were isolated from the MLNs after euthanasia on day 23. (**A**,**B**) Proportion of PEDV-positive DCs. (**C**) Representative flow cytometry plot of the DC subset distribution. (**D**) Frequencies of PEDV-positive DC subsets. Data are presented as the mean ± SD. * *p* < 0.05, ** *p* < 0.01. Unpaired Student’s *t* test.

**Figure 7 vaccines-10-01810-f007:**
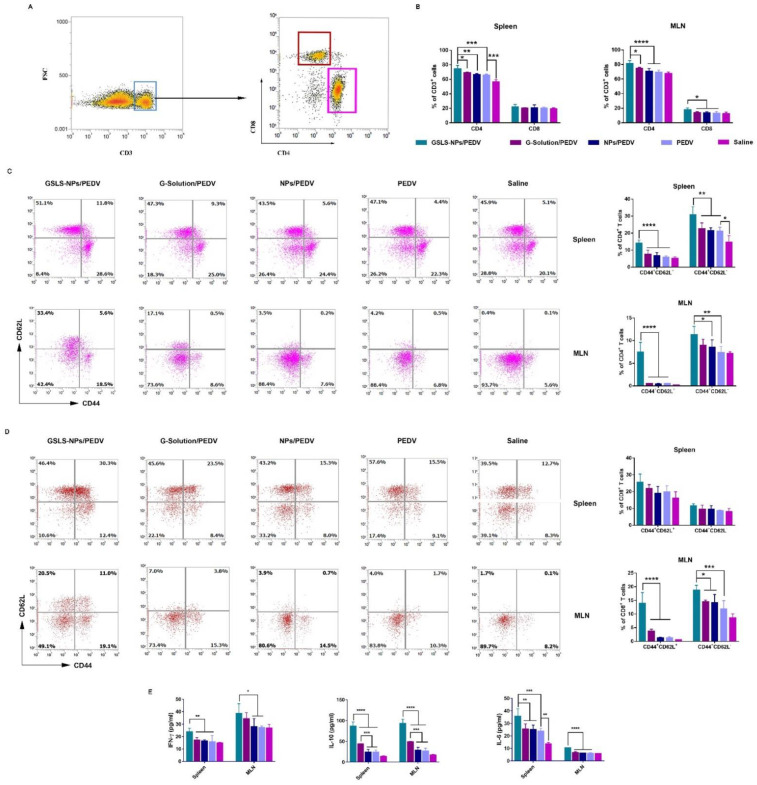
GSLS-NPs expanded PEDV-specific T-cell responses in the spleen and MLNs. Mice (n = 6) were gavaged and immunized as described above, and mononuclear cells were isolated from the spleen and MLNs after euthanasia on day 35. (**A**,**B**) Frequencies of CD4^+^ and CD8^+^ T cells. (**C**,**D**) Representative dot plots and percentages of memory (CD44^hi^ CD62L^hi^) and effector (CD44^hi^ CD62L^lo^) CD4^+^/CD8^+^ T cells. (**E**) Cytokine production in response to re-stimulation with the PEDV antigen. Data are presented as the mean ± SD. Statistical analysis was performed using an unpaired Student’s *t* test. * *p* < 0.05, ** *p* < 0.01, *** *p* < *0*.001, **** *p* < 0.0001.

**Figure 8 vaccines-10-01810-f008:**
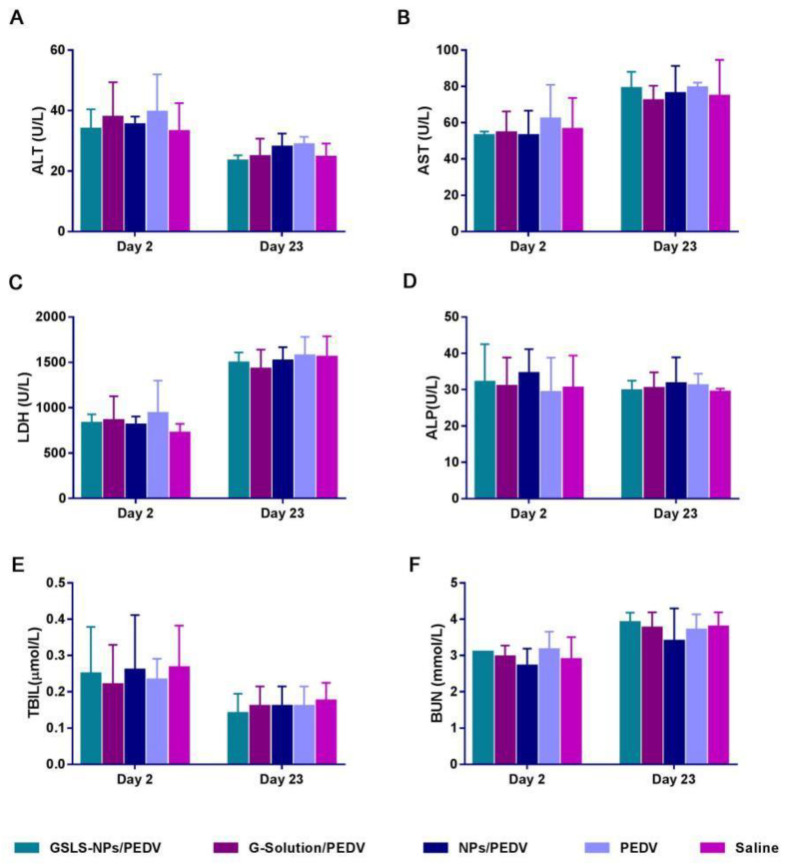
Serum levels of alanine aminotransferase (ALT), aspartate aminotransferase (AST), lactate dehydrogenase (LDH), alkaline phosphatase (ALP), total bilirubin (TBIL) and blood urea nitrogen (BUN). Mice (n = 6) were gavaged and immunized as described above, and serum was collected on days 2 and 23. Data are presented as the mean ± SD. Statistical differences were determined by an unpaired Student’s *t* test.

## Data Availability

All data generated or analyzed during the current study are available from the corresponding author on reasonable request.

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
