# Peer review of "Immune Enhancement of Nanoparticle-Encapsulated Ginseng Stem-Leaf Saponins on Porcine Epidemic Diarrhea Virus Vaccine in Mice"

_vaccines, 2022, doi:10.3390/vaccines10111810_

Round 1
Reviewer 1 Report
1. In the title, the authors should mention about the model animal (mice).
2. Line 31: “Attempts to develop a safe and effective vaccine have been unsuccessful”. Actually, in some pig farms, some commercial vaccines could provide partial protection against PEDV. Please be more accurate.
3. In conclusion, the optimal concentration for GSLS-NPs in mice model should be stipulated.
Author Response
- In the title, the authors should mention about the model animal (mice).
A : Line 4. “ in mice” has been added to the title.
- Line 31: “Attempts to develop a safe and effective vaccine have been unsuccessful”. Actually, in some pig farms, some commercial vaccines could provide partial protection against PEDV. Please be more accurate.
A: Line 15. “Attempts to develop a safe and effective vaccine have been unsuccessful” has been changed to “Current vaccines do not provide complete protection against PEDV”.
- In conclusion, the optimal concentration for GSLS-NPs in mice model should be stipulated.
A: Line 624-625. “at the dose of 5 mg/kg” has been added.
Reviewer 2 Report
In this study Su et al. developed a GSLS-PLGA nanoparticles and evaluated the mucosal adjuvant efficacy in vitro and in vivo for improving the efficacy of parenteral vaccines against PEDV. The work is important in the field, although mice are not experimental model for studying PEDV infection.
Major comments
1. For cytotoxicity study on DC2.4 cells, authors used LPS as a positive control. However, LPS is not a correct positive control for cytotoxicity. LPS is generally used as positive control for proliferation. It is reflected in their result (Fig 2A), where LPS treated cells showing slightly higher % of viable cells. Authors should have used SDS or Triton-X 100 as positive control to show the lysis of cells.
2. From the methods it is not clear how was the antigen uptake ratio was calculated. Also, why there is about 20% uptake ratio in NC, where there is no antigen added (Fig 2C).
3. To show enhanced PEDV specific antibody response by GSLS-NPs authors should have determined the titer of antibody in different groups by serial dilution of serum instead of using ELISA OD values from single dilution.
4. Authors used immunohistochemistry to identify IgA secreting cells in the intestine. Since this method only detect total IgA positive cells and not PEDV specific IgA secreting cells, authors should have used additional control GSLS-NPs alone without PEDV.
5, For neutralization assay in the methods section, please briefly explain how the titers were calculated.
6. In both figures 5 and 6, there is significant % of PEDV positive DCs in spleen and MLN of saline control group. There should not be any PEDV positive DCs in this group. Presence of PEDV positive DCs in the saline group indicates that there is nonspecific binding of PEDV antibody. Authors need to re-analyze these flow cytometry data.
Author Response
- For cytotoxicity study on DC2.4 cells, authors used LPS as a positive control. However, LPS is not a correct positive control for cytotoxicity. LPS is generally used as positive control for proliferation. It is reflected in their result (Fig 2A), where LPS treated cells showing slightly higher % of viable cells. Authors should have used SDS or Triton-X 100 as positive control to show the lysis of cells.
A: LPS is an agonist of immune cells and also has cytotoxic effect, depending on the dose and duration of action. The purpose of studying the proliferative or toxic effects of drugs on cells is the same, which is to understand the effect of drugs on cell vitality. Therefore, I think LPS can be used as a positive control for cell proliferation test, and can also be used as a positive control for cytotoxicity test.
- From the methods it is not clear how was the antigen uptake ratio was calculated. Also, why there is about 20% uptake ratio in NC, where there is no antigen added (Fig 2C).
A: The uptake ratio is the mean value of the percentages of PEDV positive cells in the total gated cells. 20% in NC might be the background value.
- To show enhanced PEDV specific antibody response by GSLS-NPs authors should have determined the titer of antibody in different groups by serial dilution of serum instead of using ELISA OD values from single dilution.
A:We optimized the sample dilution in the preliminary test.
- Authors used immunohistochemistry to identify IgA secreting cells in the intestine. Since this method only detect total IgA positive cells and not PEDV specific IgA secreting cells, authors should have used additional control GSLS-NPs alone without PEDV.
A:In this study, the immunohistochemistry test was used to understand the adjuvant effect of GSLS-NPs on IgA-secreting cells in small intestine. The PEDV vaccine alone immunized group was set as a control group, and the adjuvant effect of GSLS-NPs could be elucidated by comparing the immunized groups with and without GSLS-NPs. In addition, the adjuvant effect of GSLS-NPs on PEDV-specific IgA antibody responses was investigated by ELISA (Fig.3E).
- For neutralization assay in the methods section, please briefly explain how the titers were calculated.
A: Line 246-248. Brief description and citation have been added.
- In both figures 5 and 6, there is significant % of PEDV positive DCs in spleen and MLN of saline control group. There should not be any PEDV positive DCs in this group. Presence of PEDV positive DCs in the saline group indicates that there is nonspecific binding of PEDV antibody. Authors need to re-analyze these flow cytometry data.
A: In flow cytometry assays, isotype controls were set up to eliminate interference from nonspecific binding, and all samples from the same batch were tested under the same conditions to exclude interference from background signals, so the data were reliable.
Reviewer 3 Report
The manuscript by Su et al. describes immunological effects of GSLS-NP in PEDV vaccination. In vitro and in vivo experiments, looking at humoral responses, dendritic cell activation, and T cell memory and effector subsets, show the potential of the compound as an adjuvant for a PEDV vaccine. The document is well written, show a comprehensive analysis of the effects of GSLS-NS effects in immune cells and immune system immune responses. The paper will certainly be of interest not only for PEDV but also for other research/vaccine development fields.
Minor concerns:
- Please review/clarify the results of cytotoxic assays (Fig 2A) regarding the positive control (LPS). Would decreased viability being expected with treatment with LPS?
- Overall figures are to small and difficult to read. I recommend presenting bigger figures or using larger fonts in the images.
Author Response
- Please review/clarify the results of cytotoxic assays (Fig 2A) regarding the positive control (LPS). Would decreased viability being expected with treatment with LPS?
A: LPS is an agonist of immune cells and also has cytotoxic effect, depending on the dose and duration of action. In our results, LPS did not reduce the number of DC2.4 cells, but slightly promoted cell proliferation, which could reflect its effect on cell viability.
- Overall figures are to small and difficult to read. I recommend presenting bigger figures or using larger fonts in the images.
A: The fonts in figures have been changed to larger ones.